# Dismantling of Reinforced Concrete Using Steam Pressure Cracking System: Drilling and Crack Propagation

**DOI:** 10.3390/ma16041398

**Published:** 2023-02-07

**Authors:** Osamu Kamiya, Mamoru Takahashi, Yasuyuki Miyano, Shinichi Ito, Kenji Murata, Makoto Kawano, Arata Maisawa, Jumpei Nanao, Takashi Kazumi, Masanobu Nakatsu, Hiroyuki Mizuma, Tatsuya Miyota, Kota Nagao, Yuichi Iwama

**Affiliations:** 1Department of System Design Engineering, Faculty of Engineering Science, Akita University, Akita 010-8502, Japan; 2Research & Development Department, Nippon Koki Co., Ltd., Shirakawa 961-8686, Japan; 3SANWA TEKKI CORPORATION, 6-4-6 Minami-Shinagawa, Shinagawa-ku, Tokyo 140-8669, Japan

**Keywords:** quick dismantling, reinforced concrete, steam pressure cracking, less vibration, remote system, non-pollution

## Abstract

This study investigated a new dismantling system for concrete structures using a steam pressure cracking agent. We improved the mechanical systems such that it can drill through reinforcing steel bars. Therefore, the control method of the system and shape of the drill tip were improved. When the drill tip is stuck with chips and stopped, it is automatically pulled out and reinserted to recover the rotation. By changing the tip angle of the drill bit from 75° to 90°, it became possible to cut reinforcing bars, which were difficult to cut previously. In addition, we designed a crawler-type mechanical system and improved it such that it can be moved to the appropriate position and operated at any angle. This study revealed that the energy required for the drilling process accounts for more than 90% of the total dismantling energy. Through experiments using an impact hammer drill and observations of fracture surfaces using a three-dimensional scanner, we analysed the characteristics of reinforced concrete. In addition, the feasibility of the design for dismantling reinforced concrete was confirmed based on the determined energy associated with crack propagation.

## 1. Introduction

Japan is prone to natural disasters [1] and technology must be applied to protect its people. In the immediate aftermath of an earthquake or typhoon, broken reinforced concrete structures must be dismantled to rescue people and restore the infrastructure. Cracking the concrete structures of nuclear power plants is one such application [2,3]. The cracking of concrete reinforced by steel at dangerous sites with high radiation concentrations [4] and contaminated water is dangerous for workers and may cause environmental pollution [5,6]. The aim of this research was to create a remote dismantling system that is safe, rapid, environmentally friendly, and consumes less energy. 

The authors previously developed a concrete dismantling system using steam pressure cracking (SPC) agents. In this study, the authors improved the mechanical system such that it can continuously drill reinforcing steel bars as well as concrete. Therefore, the control method of the system and shape of the drill tip were improved. In addition, we designed a crawler-type mechanical system and improved it such that it can be moved to the appropriate position and operated at any angle. 

The SPC agent was developed by Nippon Koki, to which five of the authors of this paper belong [7,8,9,10]. The reaction pressure of the SPC agent is one-tenth that of normal explosive; therefore, the agent can be safely used in residential areas. The agent was used to dismantle a concrete waste structure on the sea shore after the tsunami caused by the Great East Japan earthquake on 11 March 2011.

The mechanical properties of industrial concrete vary depending on manufacturing conditions, curing time, and usage environment [11,12,13]. Concrete is a mixture of cement, water, and sand, and the mixing ratio of these materials affects its strength [13]. The higher the sand ratio and lower the amount of water, the greater the drying density of the concrete and the greater the compressive strength. In addition, the hydration reaction when concrete solidifies is affected by the temperature, humidity and curing time [12]. As a result, the compressive strength of concrete varies significantly from 5 to 70 MPa [13]. In this study, concrete with strengths from 20 to 50 MPa was used. 

The concrete used in nuclear power plants has the same basic components as conventional concrete, but more recently, high-performance concrete (HPC) has been developed and used [14]. HPC has a high strength and durability. These properties are achieved through microstructural improvement, water content reduction, and controlled curing. For example, HPC with a strength of 64.5 MPa was used in the Civaux-2 nuclear power plant of France [14]. Furthermore, for a better shielding performance of radioactivity in the concrete near the reactor core, iron oxide is mixed to increase the density of the concrete. The large reinforced concrete used in this study was assumed to have a reactor structure with a strength of 50 MPa. 

Methods for dismantling concrete are divided into mechanical, thermal, and chemical methods. Mechanical dismantling methods include hydraulic breakers, ball and crane, saw cutter and abrasive water jets [15]. They are heavy machine systems that are difficult to use in nuclear power plants. Thermal dismantling, developed by Takeuchi et al. [16], is the plasma cutting for concrete plate. A concrete plate with a thickness of 200 mm can be cut using the plasma current of 530 A current. It may be difficult to use in the field, because it requires a moving device while supplying plasma-consumable electrodes. There are three types of chemical dismantling methods which include SPC in this study. The first category is explosives such as water–gel explosives [15], which generate large vibrations and smaller particles of industrial waste compared with SPC [8]. These particles pollute the environment and cannot be used to dismantle nuclear reactors. 

The second type is a static chemical reaction agent that can produce high pressure [17]. The chemical expands in the enclosed space and creates sufficient pressure to safely crack the concrete. However, it takes a long time, up to several hours; therefore, it cannot be used for the emergency demolition of concrete. Natanzi demonstrated the use of soundless chemical demolition agents (SCDAs) applied to the construction of a subway platform in New York City. She indicated that the minimum demolition time was approximately 24 h [18].

The authors created a dismantling system using an SPC agent [19,20]. In this study, we investigated the drilling of concrete as a preparatory process, which has not been investigated until now. Additionally, a new method using a scanner was examined to analyse the fracture surface after cracking and is presented in this study. 

The authors previously studied the moment of cracking using SPC [21,22]. This study clarified that over 90% of the dismantling time and energy were consumed during the drilling process before cracking. In addition, a fracture surface was observed after crushing. In this study, the authors drilled concrete, cut reinforcing steel bars, and analysed the fracture surface after fracturing. Subsequently, by changing the mechanism of the machine and the shape of the bit tip, we attempted to improve drill efficiency. Therefore, we improved the efficiency of the dismantling system. 

We propose a system that can safely and quickly dismantle industrial concrete waste generated during disasters and emergencies. We also considered the overall energy consumption of the demolition process when using the SPC system.

This research was conducted based on an official tripartite agreement with Akita University, Nippon Koki Co. Ltd. and SANWA TEKKI CORPORATION.

## 2. Materials and Methods

### 2.1. Materials Used

The mechanical properties of the concrete used in this study are presented in Table 1. The concrete was produced by mixing cement, sand, aggregate stone, and water. As mentioned in the Introduction, the strength of concrete varies depending on the mixing ratio and curing time. In this study, assuming high-strength concrete for a reactor structure, the compressive strength of a large test piece (1 m × 1 m × 0.5 m) was adjusted to approximately 50 MPa. The volume fraction was 1:2:4:0.5, and the mixture was cured for 3 weeks after mixing. The concrete was reinforced using steel rebar with a diameter of 9 or 12 mm. The steel rebars used were SD345 (JIS: Japanese Industrial Standard) with tensile strength of 490 MPa.

The chemical composition and physical properties of the SPC applied in this study are presented in Table 2 and Table 3, respectively. The ignition temperature of the SPC agent is 793 K (520 °C). The chemical reaction of SPC, as shown in Equation (1), occurs at an ignition temperature that produces water and heat. The SPC reaction speed of 300 [m/s] in Table 3 is lower than that of a water–gel explosive of 5800 m/s [23]. The lower reaction speed indicates that SPC generates a mild reaction compared with a general explosive.
(1)$$2\mathrm{Al}+6\mathrm{CuO}+\mathrm{KAl}{\left({\mathrm{SO}}_{4}\right)}_{2}\cdot 12H_{2}O\;\to {\;\mathrm{Al}}_{2}O_{3}+3{\mathrm{Cu}}_{2}O+\mathrm{KAl}{\left({\mathrm{SO}}_{4}\right)}_{2}+12H_{2}O↑+1170\;\mathrm{kJ}/\mathrm{kg}$$

The water instantly evaporates into steam and produces high pressure within an enclosed area. This is a type of thermite reaction (Goldschmidt reaction). Therefore, the generated pressures can be used for concrete cracking. The pre-experimental results are shown in Figure 1. Cracking was suitably controlled using induction holes, which were analysed in a previous study by the authors. The theoretical background of the controlled cracking is described in the next section.

### 2.2. Fractography Method of Concrete Surface Cracking Using SPC 

The authors measured the macro roughness [mm] of the cracked concrete cracked surface using three dimensional scanners of POP 3D Scanner. (Revopointe 3D Technologies Inc. Shanghai, China). The accuracy in the depth direction was 0.05 mm. Figure 2 shows the scanning data for the concrete surface. Macro roughness was defined as the maximum values within the standard square area. In this study, an area of 30 mm × 30 mm was used. The stereolithography(STL) data measured by the 3D scanner (https://www.revopoint3d.com/download/, accessed on 4 February 2023) were analysed using the application FlashPrint (https://www.flashforge.com/download-center/63, accessed on 4 February 2023) (Flashforge Corporation, Zhenjiang, China) as shown in Figure 2. The analysis method of roughness at the cracked concreate surface was as follows:(1)Scanning the concrete surface using a precise 3D scanner (Revopint 3D)(2)Changing the scanning date to STL data using scanner software.(3)Reading STL data using 3D-printing software (FlashPrint).(4)Analysing the roughness data (z-direction) from the 3D coordinate data in Figure 2.

### 2.3. Design of Remote Control SPC System

The machine of the remote-control SPC system is shown in Figure 3. The function of this machine is to drill holes in the concrete and insert SPC capsules into the hole. The energy consumption for drilling concrete is higher than that for the SPC reaction. Improvements in the drilling process lead to a high efficiency of the entire dismantling process. The control method and drill tip geometry of the previous SPC system [10,20,21,22] were improved in this study. First, when the drill tip is stuck with chips and stopped, it is automatically pulled out and reinserted to recover the rotation. Second, the tip angle of 75° changes to 90°, as shown in Figure 3a,b, to cut steel bars and not only concrete. 

The system must be operated in extreme environments with high radiation level and in water. Therefore, to avoid failure, the system is operated by supplying only air pressure. In this study, the supplied air was at 0.5 MPa and had a flow velocity of 0.85 m^3^/min (0.014 m^3^/s), which maintained the power of the impact drilling machine at 2.24 kW. 

Electrical ICs are affected by weakness when exposed to radiation [4,24]. Therefore, applying a robotic system in this high-density radiation environment is difficult. Studies have shown that semiconductors are damaged by the effects of displacement damage (DD), total ionizing dose (TID), and of single event effects (SEE) under high radiation densities.

The advantage of this system is that it is driven by air pressure without a semiconductor circuit, making it is stable in extreme environments, such as severe radiation, water, and high temperature. 

The standard dismantling process using the SPC system is as follows [18].

(1)Complete planning for dismantling.(2)Design of the cracking order for the entire structure.(3)Positioning using a precision crane.(4)Drilling using the remote-control SPC system.(5)Inserting of the SPC cartridge(6)Sealing the hole with glue(7)Wiring the SPC(8)Confirmation of the wire connection(9)Using electrical ignition(10)Confirmation of total cracking.

Operations (4) to (7) can be performed automatically. Processes of (1), (2), (3), (8), (9), and (10) are manually performed by an engineer. 

As shown in Figure 4a, the SPC system was set up using a precision crane and operating under the conditions listed in Table 4. A root-hammer was used in the SPC system to drill holes into the concrete. The concrete can be drilled smoothly, but steel bar drilling would slow down and almost stop. In this study, the drilling process and concrete and steel bars were examined in detail to obtain a solution [21,22].

As shown in Figure 4b, the large concrete test piece has an area of 1000 mm^2^ and thickness of 500 mm. The concrete had eight holes with a diameter of 34 mm and a depth of 270 mm. The central four holes were SPC holes, and the peripheral four holes were induction holes. The concrete was cracked using the standard dismantling process described in steps (3) to (10).

One advantage of this system is the use of a drill bit to insert the SPC capsule, as shown in Figure 5a,b. After the hole is drilled, the remote hand grasps the SPC capsule (Figure 5a), rotates 180°, and inserts into the hole (Figure 5b). Using this analogue mechanism without a semiconductor sensor, the system can still insert the SPC capsules into the holes. The system can be used sustainably under high radiation conditions. General robotics systems with semiconductors would be affected by the radiation effect [4].

The authors designed a crawler type SPC system corresponding to actual use in a flat area, as shown in Figure 6 and Table 5. The basic mechanism and control method of the crawler are the same as those of the tripod-type. The crawler machine is set by a crane close to the actual site. The crawler system moves to the exact position using two caterpillars, sets the angle of the air drill, fine-tunes the bit tip while observing with a fibre camera, and then drills the concrete and inserts the SPC capsule. In an emergency, it can be immediately retrieved using a crane.

## 3. Results and Discussions

### 3.1. Drilling of Reinforced Concretes

The authors examined the drilling of reinforced concrete using root-hammers in a steam pressure cracking system as shown in Figure 7a.

The reinforced concrete was drilled using the bit shown in Figure 7b. The drilling bit was operated under the following conditions: a press load of 76.4 [N] and drilling speed in concrete of 0.6 [mm/s] (Figure 8). This machine bit can cut the reinforcement steel rebars in the concrete, as shown in Figure 7c. The drilling speed in the steel rebar is lower than that in concrete under the conditions of φ42 bit and φ9 steel rebar diameters. Initially, a tip angle of 75° was used, but the iron rod was plastically deformed horizontally and did not lead to cutting. Therefore, when the tip angle was set to 90°, the lateral plastic deformation was reduced and it was possible to cut. Therefore, the experiment was continued at 90°.

The relationship between the distance of the drill tip from the surface and drilling time while the SPC system was operating is shown in Figure 8. The straight line from the origin indicates that the tip of the bit contacted the surface of the concrete and began drilling. The gradient of the line indicates the drilling speed of concrete cutting. The gradient suddenly changed at a drilling distance of 35 mm, indicating that the drill tip reached the reinforced steel bar. The drilling speeds of concrete and steel were 0.61 and 0.0175 mm/s, respectively. The drilling speed recovered at a drilling distance of 45 mm, because the steel bar of 9 mm diameter was drilled through and the concrete was being drilled again. 

In the drilling process, reinforcing bars require more energy than concrete. The energy consumption can be calculated using the power of the root-hammer of this system as follows:

The results are analysed here in terms of the material removal rate (MRR) in alignment with the standard theory for the drilling process.
(2)$$MRR\;in\;Drilling=\left(\frac{\pi D^{2}}{4}\right)fN$$
where *MRR* [mm^3^/s] is the material removal rate, *D* [mm] is the drill diameter, f [mm/rev] is the feed per revolution and *N* is the rotational speed of the bit per second. Equation (2) can be used for continuous cutting and drilling. For discontinuous drilling, the average value can be determined using the following equation:*MRR* = *Vr/Tr*
(3)
where *Vr* is the chip volume [mm^3^] when drilling at time *Tr* [s]. Both *Vr* and *Tr* can be calculated using the experimental data presented in Figure 8.

The MRR of concrete and steel are *MRRc* = 844 mm^3^/s and *MRRs* = 3.66 mm^3^/s, respectively. 

The material constant of the specific energy consumption (*E_s_*) [J/mm^3^] can be calculated using the following equation,
*E_s_* = *P*/*MRR*
(4)
where *P* is the power [W] or [J/s] of the machine system.

The power of the root-hammer in the steam pressure cracking system is 2.24 kW (2240 J/s). The specific energy of concrete (*E_s_*)_c_ and steel (*E_s_*)_s_ are 2.65 and 612.0 [J/mm^3^], respectively. These data enabled the calculations of the total energy consumption (*E_total_)* based on the size of the holes in the of concrete.
(5)$$E_{total}=V_{r}E_{s}$$

The energy consumption of drilling is not negligible in a dismantling system. In the next section, the drilling energy is compared with the SPC energy. 

### 3.2. Analytical Fractography of Concrete

In terms of the SPC energy, initially, the compressive elastic energy propagated in the concrete brick and caused damage, such as microcracks. However, this did not lead to final destruction. Thereafter, the steam pressure reached its maximum, and a final rupture occurred on the straight line connecting the induction hole and the SPC centre hole. The authors consider that the main SPC energy was absorbed at the surface of the concrete. Based on the roughness of the concrete fracture surface, we observed that the distribution of the surface roughness is not uniform. 

Figure 9 shows that the roughness of the area close to the SPC centre was smaller than that further from the centre. As shown in Figure 9a, the roughness of the general square surface (70 mm × 70 mm) was as high as 12.3 mm. This was because the aggregate separated at the boundary of the cement matrix indicated a large amount of energy. In contrast, as shown in Figure 9b, the roughness of the centre area was as small as 5.9 mm, because the brittle aggregate fracture inside indicates a low energy absorption by the surface. The authors consider that the velocity of crack propagation in the centre was high because of the high SPC pressure applied to the brittle fractures with low energy.

The variation in the roughness shown in Figure 9c can be related to the difference in the crack propagation speed. Generally, the failure mode of the material exhibits a tendency to brittle fracturing with an impact load compared with a static load. The impact pressure owing to the SPC was the highest near the centre, and the pressure decreased as it moved away. That is, the closer to the centre, the more the material become brittle, and the cracks propagate with brittleness and flatly.

A detailed observation of the regions with different roughness values is shown in Figure 10. As shown in Figure 10a, in the region away from the SPC, the roughness was large because cracks often propagate through a boundary between the aggregate and the cement, which is called intergranular fracture. In contrast, in the region close to the SPC, as shown in Figure 10b, cracks mostly propagate in the aggregate, which is called transgranular fracturing; the fracture surface is flat, and the roughness is low. The reason for this is considered to be the concrete embrittlement because the strain rate is high near the SPC. To dismantle with a small amount of SPC agent, it is better to propagate the crack flatly using intragranular fracturing to control the crack; the crack can easily stop the resistance by breaking the grain boundaries. Whether cracks penetrate the aggregate or propagate along the boundary is an important problem for future research and requires theoretical analysis [25,26,27]. In the future, an appropriate strength should be achieved at the interface surface, and concrete with high strength and easy fracture control is desired.

Figure 11a,b show the drilling energy (D) and SPC energy that were used to dismantle the reinforced concrete. 

Here, we discuss the SPC energy absorption by the surface energy from the perspective of fracture mechanics. Figure 11b shows the relationship between the SPC energy (W) and crack surface area (S).

The compressible pressure energy caused by the SPC agent is converted into strain energy, W, which is released through crack propagation over length c. The fracture energy can be understood as the surface energy γ_s_ of the cracked surface. This value can be expressed by the following equation: W = 2cγ_s_(6)
where W is the overall strain energy corresponding to the energy produced by the SPC agent, c is the crack propagation length per unit depth corresponding to the area of the fracture surface, and γ_s_ is the surface energy of the reinforced concrete used. The authors consider that the value of γ_s_ includes the damage to the concrete as micro cracks that are initiated by the elastic tensile stress wave. The energy of the SPC required to form a 1 m^2^ crack in the concrete dismantled is 276.7 kJ/m^2^, corresponding to γ_s_ in Equation (6).
W = 276.7s [kJ] (7)
where s is the area of the fracture surface.

The value of the surface energy γ_s_ is considered to include some energy losses, as indicated in Table 6. The value of γ_s_ can be considered nominal, that is, the true value of the surface energy can be obtained by subtracting the various energy losses shown in Table 6 from this nominal value. Part of the reaction energy of the SPC leaks through the gap of the main crack and is lost. Immediately after the reaction, energy loss occurs owing to the propagation of the elastic waves. In addition, many secondary cracks, which differ from the main cracks, occur inside the concrete and consume energy. However, it is difficult to determine these energy losses and the true surface energy γ_t_, both experimentally and theoretically. The purpose of this study was to design a dismantling system based on the nominal surface energy.

The drilling energy of concrete and SPC energy as the dismantling energy consumption is depicted in Figure 11a. The cutting surface consisted of a fracture area and drilled holes. The drilled hole area ratio was small as shown in Figure 9. However, the calculated energy consumption for drilling was large, as shown in Figure 11.

The effect of the reinforcing steel bar depended on the cutting method used. The impact value when the general steel bar was broken by SPC was approximately 100 J/cm^2^ [10]. It required approximately 64 J of energy to impact cutting with a diameter of 9 [mm]. In the case of drilling a 9 mm steel bar, the energy consumption is 1634 kJ which is calculated using Equation (5). The steel bar can be cut using the system; however, the energy loss is large based on a comparison in terms of the SPC energy. 

The concept of the total dismantling energy using an SPC system is shown in Figure 12. The dismantling energy consists of drilling, elastic waves, and the SPC reaction energy. The drilling energy is calculated from the specific energy consumption (*E_s_*) and the material removal volume (Vr) (=west chip volume) using a formula of $E_{total}=V_{r}E_{s}$. The elastic wave energy controls the crack direction, and the energy is small. The required amount of SPC corresponding to crack propagation can be calculated using the surface energy (γ_s_) and the required crack area (s) using the formula W = γ_s_ s.

In this study, for the first time we experimented with the drilling process, and observed that more than 90% of the dismantling energy was input for drilling. The problem of a sudden slowdown in the machining speed when the drill tip encounters a steel bar was discovered and has not yet been resolved. From a practical perspective, an efficient drilling process is required. 

## 4. Conclusions

The authors developed a dismantling system using a steam pressure cracking (SPC) agent for a reinforced concrete structure. We examined the drilling process and a method used to observe the fractured surface.

(1)A system using an SPC agent can dismantle concrete with almost no vibration, dust, or pollution. By utilizing this feature, the authors designed a new crawler-type practical demolition system.(2)The concrete fracture surfaces were observed using a three dimensional scanner. The roughness close to the SPC centre was smaller than that in the other areas, because trans-aggregate stone fracture occurred in the SPC centre area. Additionally, the aggregate brittle fracture inside indicates low energy absorption by the surface.(3)The specific energy of the concrete was measured during drilling. The energy consumption of drilling can be estimated from the perspective of the overall dismantling process. By increasing the angle of the drill tip, the drill energy tended to decrease. By changing the drill tip angle from 75° to 90°, it became possible to cut reinforcing bars, which were difficult to cut previously.(4)Steel bars in reinforced concrete can be easily obtained using SPC energy. However, in the drilling process, steel requires a large amount of high energy for cutting.(5)The energy consumption of drilling and SPC per unit of dismantling surface examined in this study enabled us to accurately estimate the required materials and time.(6)This study revealed that the energy required for the drilling process accounts for more than 90% of the total dismantling energy.

A future task is to improve the efficiency of the drilling process by optimising the bit shape and machine operating conditions.

## 5. Patents

The authors hold the following patent related to this study.

Invented by Kamiya O., Murata K., Mizuma H., Iwama Y., Nakatsu M., Cracking Agent insert Machine System. Published patent gazette(A), Japan Patent Office: Patent number 2021-148367.

## Figures and Tables

**Figure 1 materials-16-01398-f001:**
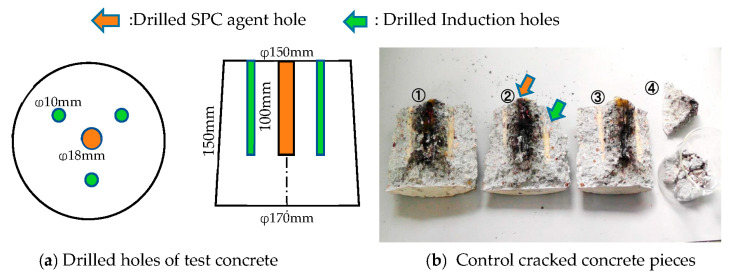
Controlled cracking of a small concrete piece (Fc = 20 MPa) using the SPC agent and induction holes drilled using an impact hammer.

**Figure 2 materials-16-01398-f002:**
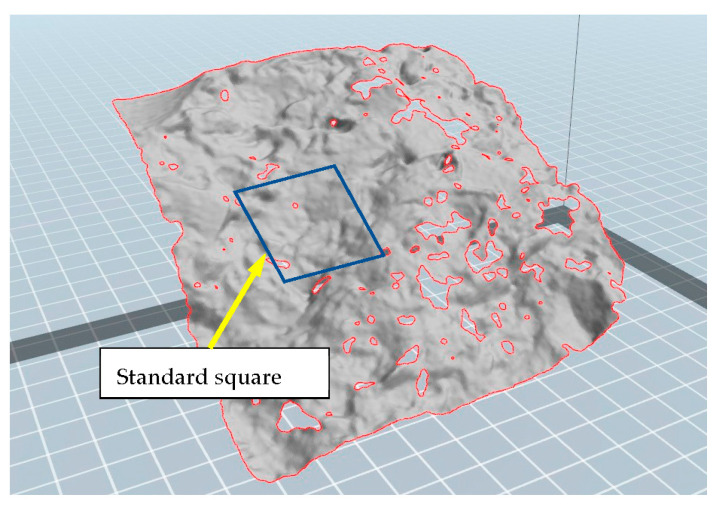
Concrete surface scanning data and the standard square area of 30 mm × 30 mm.

**Figure 3 materials-16-01398-f003:**
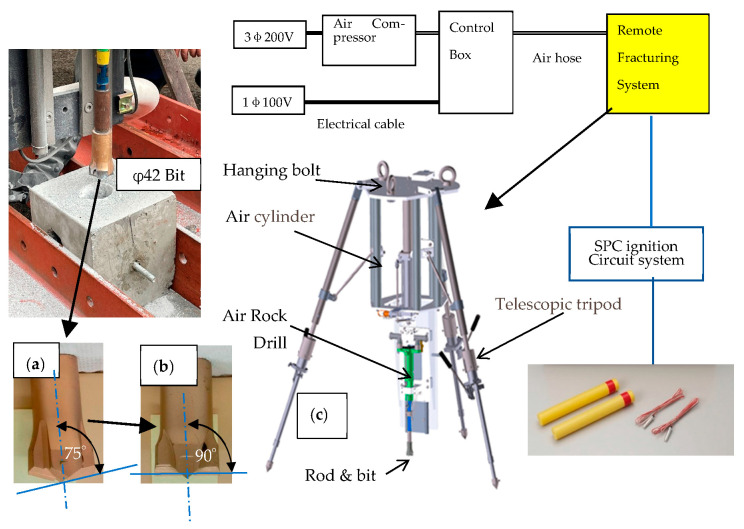
Complete design of the remote-control cracking system using SPC. Preparations of two types of drilling bit. (**a**) 75° of tip angle for concrete only, (**b**) 90° tip angel for both concrete and steel bars and (**c**) tripod type SPC system.

**Figure 4 materials-16-01398-f004:**
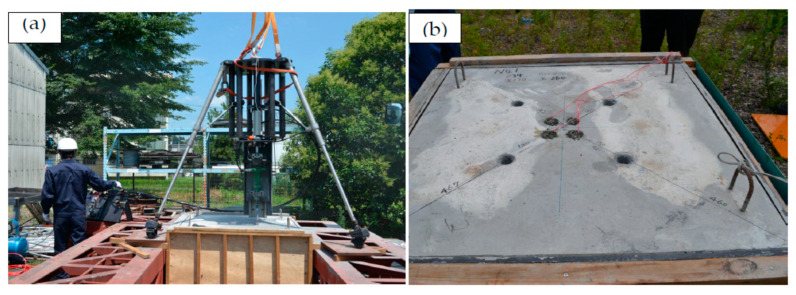
(**a**) Setting of remote cutting system and (**b**) reinforced concrete (Fc = 50 MPa). test piece (1000 mm × 1000 mm × 500 mm) with 8 drilled holes (φ34 × 270 mm). Centre 4 holes have 60 g SPC each and another hole are induction. Total drilling length is 2160 mm [10].

**Figure 5 materials-16-01398-f005:**
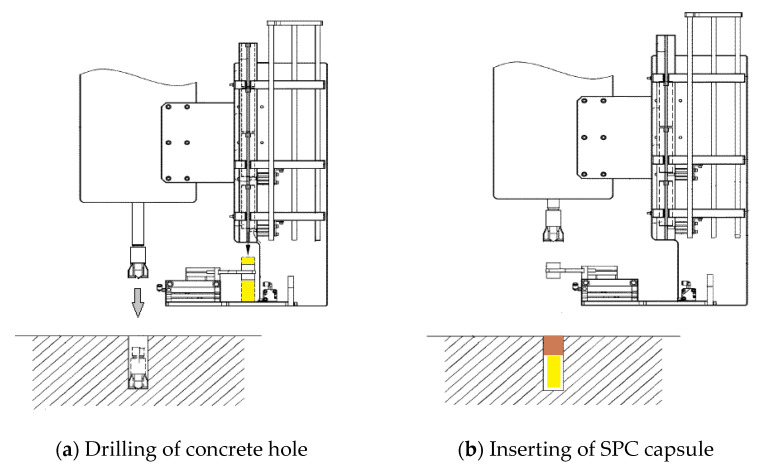
Mechanism for drilling and inserting SPC capsules into concrete holes. (**a**) drilling the hole by moving the bit in the direction of the arrow with the remote hand holding the SPC capsule, and (**b**) the remote hand turning 180°to insert SPC capsule in the hole [10].

**Figure 6 materials-16-01398-f006:**
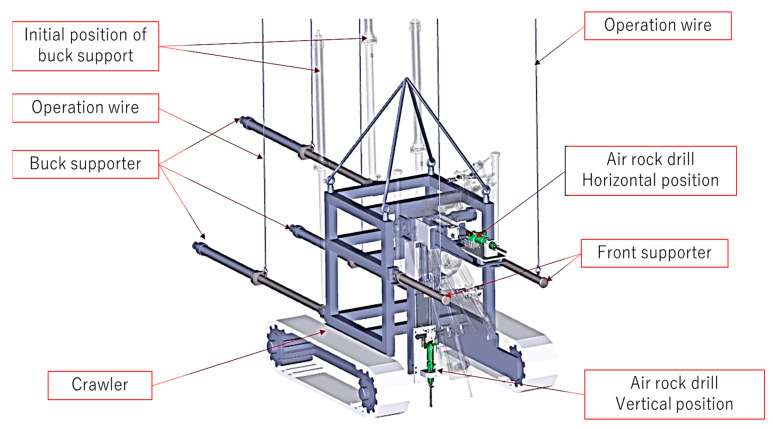
Crawler type SPC system. The crawler is set by the crane on the site. The crawler can rotate and move horizontally. The air drill angle can be adjusted stepless from vertical to horizontal.

**Figure 7 materials-16-01398-f007:**
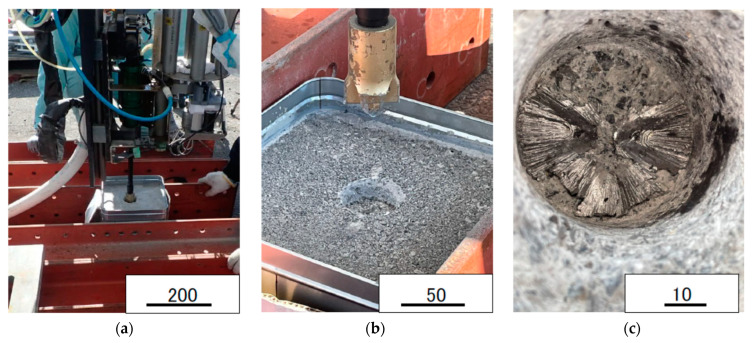
Drilling test of reinforced concrete using a root hammer, (**a**) drill initiation, (**b**) concrete waste chips resulting from drilling, and (**c**) steel bar drilled at the bottom of the hole using a 90° bit tip. It is difficult to use a 75° bit tip angle shown in Figure 5a. (Fc = 50 MPa).

**Figure 8 materials-16-01398-f008:**
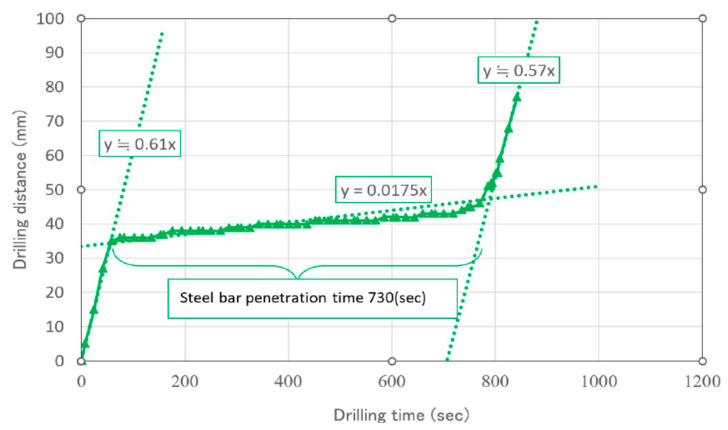
Drilling speed vc (=distance/time) in steel bars to be lower than in concrete.

**Figure 9 materials-16-01398-f009:**
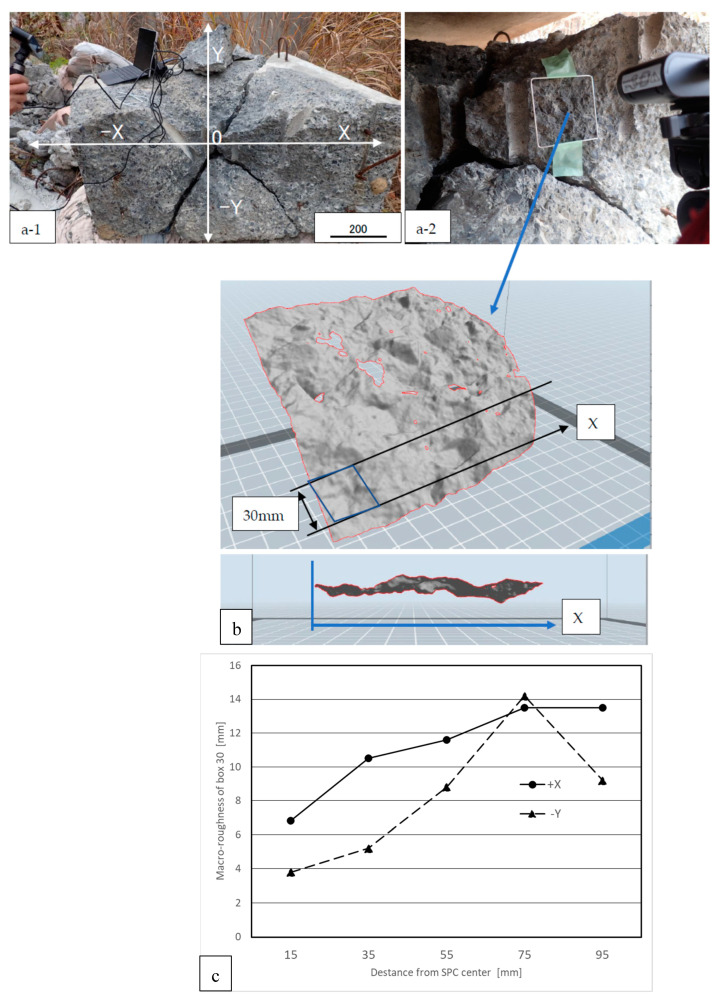
Macroscopic roughness data using a 3D scanner. (**a-1**,**a-2**) Coordinates of the concrete fracture surface. The origin is the center of SPC. (**b**) Measurements of the maximum thickness in a 30 mm × 30 mm square. (**c**) Small roughness at the centre.

**Figure 10 materials-16-01398-f010:**
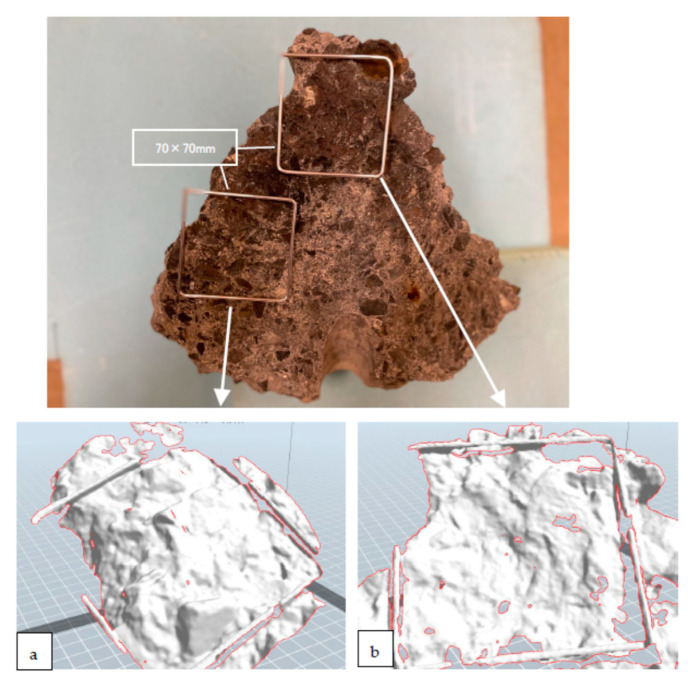
Three-dimensional observation of the concrete fracture surface. (**a**) In typical surface area, showing mixture of intergranular and transgranular surface, (**b**) in an area close to the SPC center, showing mostly transgranular and flat surface.

**Figure 11 materials-16-01398-f011:**
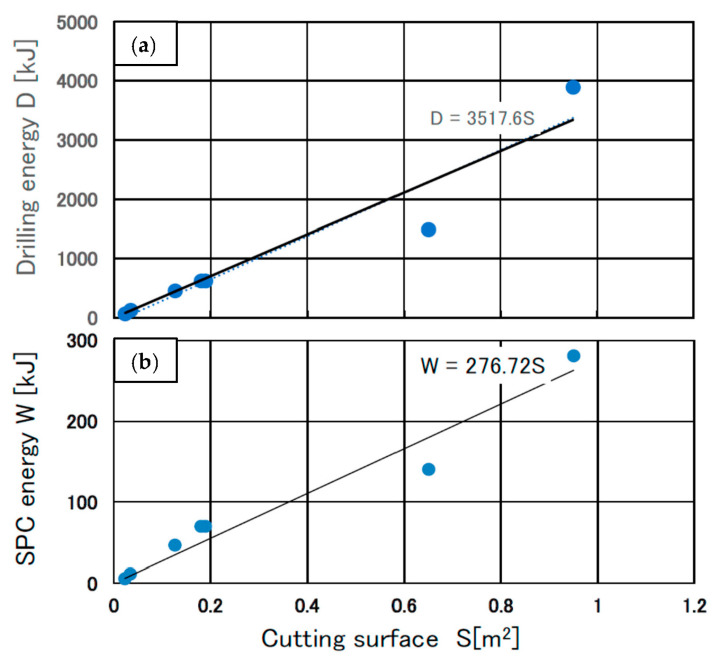
Drilling energy (D) and SPC energy (W) used to dismantle the reinforced concrete. (**a**) Relationship between D and the cutting surface (S), (**b**) relationship between W and S [10].

**Figure 12 materials-16-01398-f012:**
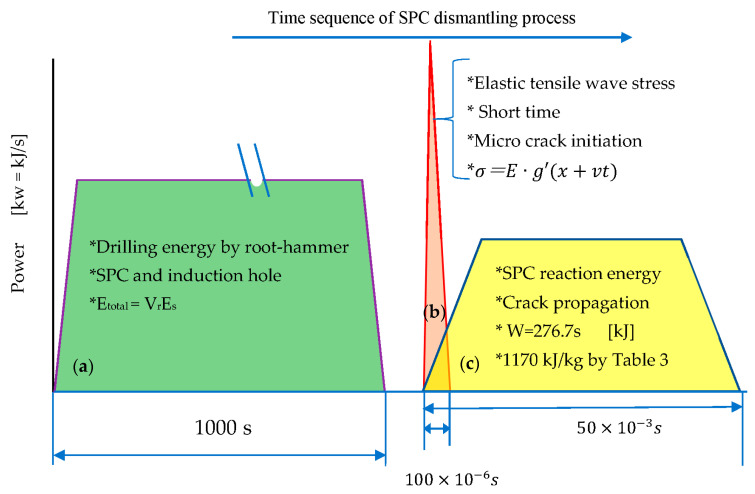
Concept of total dismantling energy using the SPC system. (**a**) Drilling: most of the energy is spent in this process, energy is the specific energy × drilling volume, (**b**) Elastic wave: the tensile stress wave may produce micro cracks, the overall energy is small. (**c**) SPC energy: the required amount of SPC can be obtained as surface energy × crack area. *: Characteristics of each energy.

**Table 1 materials-16-01398-t001:** Mean mechanical properties of concrete used.

Material Properties	Value	[Unit]
Compressive strength, F_c_	20–50	[N/mm^2^]
Tensile strength, F_t_	|F_t_| < |F_c_/10|	[N/mm^2^]
Density, *ρ*	2.3–2.5	[g/cm^3^]
Elastic modulus, E	10–35	[GPa]
Schmidt values	10–60	[R] *
Velocity of elastic wave	4500–5400	[m/s] *

*: Values measured by the author.

**Table 2 materials-16-01398-t002:** Chemical composition of the SPC agent [10].

Composition Materials	Mass %
Alum (nKAl(SO_4_)_2_12·H_2_O)	50
Copper (II) oxide (CuO)	38
Al fine powder	11 *^1^
Binder	1.0 *^2^

*^1^: Spherical aluminum particle with several microns; *^2^: Metallic soaps such as sodium stearate.

**Table 3 materials-16-01398-t003:** Physical and chemical properties of the SPC [10].

Ignition point	793 K (520 °C) or higher
Reaction speed	Less than 300 [m/s]
Rise time to maximum pressure	30–50 [10^−3^ s]
Sealed combustion pressure	300 MPa
Volume of gas produced	330 L/kg
Theoretical energy product with standard mixture	1170 kJ/kg

**Table 4 materials-16-01398-t004:** Design specifications of the dismantling system using SPC [10].

Type of Machine System	Tripod Type
Dimension	W1524 × D1319 × H1520 [mm]
Weight	250 [kgf]
Drilling stroke	250 [mm]
Bit type	Φ34, 42 [mm]
Pressing force	0~200 [N]
Drilling performance	800~1000 [mm^3^/s]
Air source pressure	0.6 [MPa]
Air consumption	0.014 [m^3^/s]
Drilling angle	75~90 [°]
Tripod stroke	400 [mm]

**Table 5 materials-16-01398-t005:** Design specifications of crawler type dismantling system using SPC.

Type of Machine System	Crawler Type
Dimension	W1650 × D2550 × H1700 [mm]
Weight	600 [kgf]
Drilling stroke	350 [mm]
Bit type	Φ34, 42 [mm]
Pressing force	0~200 [N]
Drilling performance	800~1000 [mm^3^/s]
Air source pressure	0.6 [MPa]
Air consumption	0.014 [m^3^/s]
Drilling angle	0~135 [°]

**Table 6 materials-16-01398-t006:** Energy balance of SPC and dismantling elements.

Total Energy W of Used SPC [kg]	Nominal Surface Energy γ_s_ in Cracking Area [m^2^]	True Surface Energy γ_t_ and Energy Losses
1170 [kJ/kg] times SPC mass M [kg] (W = 1170 × M)	γ_s_ [kJ/m^2^] times Cracking area S [m^2^] (W = γ_s_ × S)	True surface energy γ_t_
		SPC outgassing energy loss
		Elastic wave energy
		Secondary cracking energy
		Kinetic energy of concrete block

## Data Availability

No new data were created in this study. Data sharing is not applicable to this article.

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
