# Peer review of "Dismantling of Reinforced Concrete Using Steam Pressure Cracking System: Drilling and Crack Propagation"

_materials, 2023, doi:10.3390/ma16041398_

Round 1

Reviewer 1 Report

Dear Authors,

thank you for a rather interesting article.

The topic of rapid analysis of reinforced concrete justified by the subsequent application of the patent approach for the removal of the structure after an earthquake is narrow, specific, but interesting - it carries with it advantages and disadvantages. 

First of all, your articles (references 12 and 13) contain quite a lot of information about your system and it is not at all clear what the novelty is in the introduction of the submitted article.

There are big doubts about the novelty of the information, the authors need to explain this. 

The introduction is generally quite short and does not include information on the concretes that are or may be used in the nuclear power plant structures you mention - these materials have significantly different properties to standard concretes and therefore need to be accounted for. 

For example, see:

10.3390/ma15062173

10.3390/ma14154288

How do you explain the large variance in concrete properties shown in Table 1?

Was there an intention to create such a large variance or why is this the case?

The conclusions don't bring much new insight with respect to your previous results in the cited papers - there is a lack of detailed and clear discussion prior to the conclusions. 

A few remarks:

- at the very end of the article is the rest of some text

- photo 4 is very strange - can't take a screenshot of the program - add English text?

- the references are not in the same format and there are a large number of errors 

Author Response

2023.1.28

Dear Reviewer, 1

    Thank you for your valuable opinions.  I answered them one by one. Please check them.

Reviewer comment 1.  First of all, your articles (references 12 and 13) contain quite a lot of information about your system and it is not at all clear what the novelty is in the introduction of the submitted article.

There are big doubts about the novelty of the information, the authors need to explain this. 

Answer 1.

   Authors additionally explain in the text about the novelty of this study as follows:

Introduction.

(Line 43-47) In this study, the authors improved the mechanical system such that it can continuously drill reinforcing steel bars as well as concrete. Therefore, the control method of the system and shape of the drill tip were improved. In addition, we designed a crawler-type mechanical system and improved it such that it can be moved to the appropriate position and operated at any angle. 

(Line 150-153) “The control method and drill tip geometry of the previous SPC system [10,20–22,24] were improved in this study.  First when the drill tip is stuck with chips and stopped, it is automatically pulled out and reinserted to recover the rotation. Second, the tip angle of 75° changes to  90°,as shown in Figure 3(a),(b), to cut steel bars and not only concrete.”

(Line 197-203), “The authors designed a crawler type SPC system corresponding to actual use in  flat area as shown in Figure 6 and Table 5.  The basic mechanism and control method of the crawler are the same as those of the tripod-type.  The crawler machine is set by a crane close to the actual site. The crawler system moves to the exact position using two caterpillars, sets the angle of the air drill, fine-tunes the bit tip while observing with a fibre camera, and then drills the concrete and inserts the SPC capsule. In an emergency, it can be immediately retrieved using a crane.”

Authors also mentioned the Figure 6 and the caption of  “Figure 6. Crawler type SPC system. The crawler is set by the crane on the site. The crawler can rotate and move horizontally. The air drill angle can be adjusted stepless from vertical to horizontal.”

Reviewer comment 2. The introduction is generally quite short and does not include information on the concretes that are or may be used in the nuclear power plant structures you mention - these materials have significantly different properties to standard concretes and therefore need to be accounted for. 

For example, see:

10.3390/ma15062173

10.3390/ma14154288

Answer 2:  Thank you for comment about shortage of our introduction. The author reflected deeply and described in detail the background, purpose, characteristics of his research and the concrete materials.  (Line 33-103). Please check out the revised text.

Reviewer comment 3: How do you explain the large variance in concrete properties shown in Table 1?

Was there an intention to create such a large variance or why is this the case?

 Answer 3:   Authors explain in the introduction about the large variation in concrete properties. Points are shown as follows;  (line53-69) “The mechanical properties of industrial concrete vary depending on manufacturing conditions, curing time, and usage environment [11–13].  Concrete is a mixture of cement, water, and sand, and the mixing ratio of these materials affects its strength[13].  The higher the sand ratio and lower the amount of water, the greater the drying density of the concrete and the greater the compressive strength. In addition, the hydration reaction when concrete solidifies is affected by the temperature, humidity and curing time[12]. As a result, the compressive strength of concrete varies significantly from 5 to 70 MPa [13].  In this study, concrete with strengths from 2 0 to 5 0 MPa was used. 

 The concrete used in nuclear power plants has the same basic components as conventional concrete, but more recently, high-performance concrete (HPC) has been developed and used[14]. HPC has a high strength and durability. These properties are achieved through microstructural improvement, water content reduction, and controlled curing.  For example, HPC with a strength of 64.5 MPa was used in the Civaux-2 nuclear power plant of France [14].    Furthermore, for a better shielding performance of radioactivity in the concrete near the reactor core, iron oxide is mixed to increase the density of the concrete.  The large, reinforced concrete used in this study was assumed to have a reactor structure with a strength of 50 MPa. ”

Authors mentioned in 2.1 Materials used (line 108-111) as follows: “As mentioned in the Introduction, the strength of concrete varies depending on the mixing ratio and curing time. In this study, assuming high-strength concrete for a reactor structure, the compressive strength of a large test piece (1m×1m×0.5 m) was adjusted to approximately 50 MPa.”

Reviewer comment 4: The conclusions don't bring much new insight with respect to your previous results in the cited papers - there is a lack of detailed and clear discussion prior to the conclusions. 

 Answer 4: The author added two new conclusions in response to the discussion of the main text. “ By utilizing this feature, the authors designed a new crawler type practical demolition system.”

“.By increasing the angle of the drill tip, the drill energy tended to decrease. By changing the drill tip angle from 75 ° to 90 °, it became possible to cut reinforcing bars, which were difficult to cut previously.”

A few remarks:

- at the very end of the article is the rest of some text

Answer: The authors checked carefully about the rest of some text.

- photo 4 is very strange - can't take a screenshot of the program - add English text?

Answer: Photo 4 (Figure 4) was changed to original fine photo.

- the references are not in the same format and there are a large number of errors 

Answer: Thank you for pointing out the number of errors in references. I checked the number and corrected.

Submission Date

18 December 2022

Date of this review

08 Jan 2023 14:52:05

Reviewer 2 Report

The reviewed manuscript concerns a very practical issue, which is the demolition of concrete or reinforced concrete elements.

1. The proposed solution should significantly facilitate the demolition of concrete elements. It would be good if the developed demolition kit could independently reach the demolition site on wheels or on caterpillar tracks.

2. The article lacks an explanation as to why the tip of the drill was shaped in this way. Have other variants of this tip been considered, which could potentially improve drilling efficiency?

3. The theoretical description of cracking in the manuscript is unclear. There is no ordering of the assumptions of the computational analysis of fracture energy. These parts of the article should be thoroughly restructured and put in order.

4. It is not commented on how significant the impact of the steel reinforcement on the fracture planes will be in this case. For comparison, in this case, the same tests should be performed for concrete with and without reinforcement. Then one could comment on the effects of reinforcement caused by steel reinforcement.

5. It would be very advisable to develop an FEM model of the tested element without and with reinforcement in the analysis of the mechanism and course of cracks. Today, such analyzes are not complicated, and a more clear explanation would explain the effects of drilling and then expanding the element from the level of the hole made.

6. In the introduction to the article, the problem of methods of demolition of concrete elements should be discussed in more detail, thus expanding the list of references.

Author Response

2023.1.28

Dear Reviewer, 2

    Thank you for your valuable opinions.  I answered them one by one. Please check them.

Reviewer comment 1. The proposed solution should significantly facilitate the demolition of concrete elements. It would be good if the developed demolition kit could independently reach the demolition site on wheels or on caterpillar tracks.

Answer 1: Thank you very much for your smart comment. The authors designed a new crawler type SPC system (with caterpillars) and added as Figure 6. The caption explains “Figure 6. Crawler type SPC system. The crawler is set by the crane on the site. The crawler can rotate and move horizontally. The air drill angle can be adjusted stepless from vertical to horizontal.”  Please check this new machine in revised manuscript. I explain in the text (line 197-203)  “The authors designed a crawler type SPC system corresponding to actual use in  flat area as shown in Figure 6 and Table 5.  The basic mechanism and control method of the crawler are the same as those of the tripod-type.  The crawler machine is set by a crane close to the actual site. The crawler system moves to the exact position using two caterpillars, sets the angle of the air drill, fine-tunes the bit tip while observing with a fibre camera, and then drills the concrete and inserts the SPC capsule. In an emergency, it can be immediately retrieved using a crane.”

Reviewer comment 2. The article lacks an explanation as to why the tip of the drill was shaped in this way. Have other variants of this tip been considered, which could potentially improve drilling efficiency?

Answer 2: Authors added the decision process of drill tip shape in main text. ( Line 215-219)  ” Initially, a tip angle of 75 ° was used, but the iron rod was plastically deformed horizontally and did not lead to cutting. Therefore, when the tip angle was set to 90 °, the lateral plastic deformation was reduced and it was possible to cut. Therefore, the experiment was continued at 90 °.”  

Figure 5(a),(b) indicate both photografhy.

We are continuing experiments in search of more efficient bit shapes.

Reviewer comment 3. The theoretical description of cracking in the manuscript is unclear. There is no ordering of the assumptions of the computational analysis of fracture energy. These parts of the article should be thoroughly restructured and put in order.

Answer 3: I understand that theoretical explanation of the crack initiation point and propagation pass is unclear in this report. It is difficult for me to clarify these problems theoretically. There are several reasons. Since I calculate the surface energy assuming that all the energy of the SPC is absorbed by the crack surface. That nominal surface energy is including various energy loss.  One is the "gas leakage loss" in which the high-pressure steam of the SPC leaks through the gaps in the cracks. In addition, it is dissipated as "elastic waves", and furthermore, energy is absorbed as many "secondary cracks". Subtracting gas loss, elastic wave energy, and secondary crack energy with in  the SPC energy will require theoretical analysis and simulation to correctly determine the surface energy. On the other hand, from a design point of view, it is more convenient to include various energy loss in the surface energy in order to determine the amount of SPC required for the structure to be dismantled. We, the engineers, create a disposal system, prepare the necessary SPC, and dismantle them.

   I explain the situation in the text using additional Table 6. ( in page 13-14, line 319-328) ” The value of the surface energy γs  is considered to include some energy losses, as indicated in Table 6.  The value of  γs can be nominal, that  is, the true value of the surface energy can be obtained by subtracting the various energy losses shown in Table 6 from this nominal value. Part of the reaction energy of the SPC leaks through the gap of the main crack and is lost. Immediately after the reaction, energy loss occurs owing to the propagation of the elastic waves. In addition, many secondary cracks, which differ from the main cracks, occur inside the concrete and consume energy. However, it is difficult to determine these energy losses and the true surface energy γt , both experimentally and theoretically. The purpose of this study was to design a dismantling system based on the nominal surface energy.”

Reviewer comment 4. It is not commented on how significant the impact of the steel reinforcement on the fracture planes will be in this case. For comparison, in this case, the same tests should be performed for concrete with and without reinforcement. Then one could comment on the effects of reinforcement caused by steel reinforcement.

Answer 4: Authors compered between reinforced concrete and without reinforcement one in reference (10).  When the concrete reinforced and without reinforce were fractured by SPC, the energy consumption are similar as shown in reference Figure 9 (10). Because the steel fractured with brittle manner and low energy by impact pressure of SPC. The energy was only 100 J/cm2. When 10 bars of 12 mm diameter are arranged in concrete 1 m2, the sum of the energy when impacting and destroying this is 3.14 kJ. This value is about 1% of the coefficient of 279.8 kJ/m2, as shown in Figure 9.(10).   On the other hand, drilling energy is  about ten times bigger than SPC.

Reviewer comment 5. It would be very advisable to develop an FEM model of the tested element without and with reinforcement in the analysis of the mechanism and course of cracks. Today, such analyses are not complicated, and a more clear explanation would explain the effects of drilling and then expanding the element from the level of the hole made.

Answer 5: Thank you very much for attractive advice. I would like to analyse some kinds of FEM as an issue for the future. I understand that the elastic-plastic strain FEM for analysing the drilling process will enables better designs of the mechanical system. Also, the elastic wave FEM analysis and the fracture mechanics FEM will reveal crack initiation and propagation phenomena.  In the future, we would like to develop and improve a new SPC system and at the same time perform FEM analysis as described in the comment.

Reviewer comment 6. In the introduction to the article, the problem of methods of demolition of concrete elements should be discussed in more detail, thus expanding the list of references.

Answer 6: Thank you very much for pointing out the shortage of the Introduction. I  discussed more detail in the Introduction (line 33-103 in revised text) about the contribution of SPC method, the variation of concrete properties, the mechanical, thermal and chemical dismantling methods.

Submission Date

18 December 2022

Date of this review

10 Jan 2023 22:59:06

Answer date 2023.1.28

Osamu Kamiya

Reviewer 3 Report

General Comment

Following the previous works from the authors related with a developed novel dismantling process for reinforced concrete using steam pressure cracking system, this manuscript mainly focusses on the examination of the drilling process, on a method to observed the fractured concrete surfaces and a study based on the specific energy consumption measured during the drilling and crack propagation to estimate the required materials and time. For this, an experimental campaign was performed. The results are presented and discussed.

The topic of the manuscript, as well as the developed novel dismantling process, is very interesting and could be used in the next future as an alternative more safer and more environment friendly dismantling technique.

I consider that in the present form the manuscript is not acceptable for publication. I made some comments in order to improve the manuscript. The authors should take the comments into account and revise their manuscript.

Specific Comment 1

The manuscript needs an extensive revision to correct the reading, tense, spelling, grammar and typos. Avoid to write in the first person of plural. Several sentences are not complete and hard to understand (the manuscript is not easy to read), and several formatting issues (including symbols and numbering of figures, among others) need also to be corrected. The authors should seek to professional help for this revision and also refer to the guide for authors. I don’t present examples because they are so many.

Specific Comment 2

Abstract

The abstract must be revised. It must summarizes the achievements of the previous works from the authors, and highlight the novelty and importance of this manuscript. In addition, the main results of this research should be also summarized.

Specific Comment 3

Keywords

Additional keywords must be added and refer to the novelty of this study.

Specific Comment 4

Introduction

The introduction section is somewhat poor and to short. The literature review must be updated and improved. In a more detailed way, in the end of the introduction the novelty and importance of this study must be clearly explained, in relation with the previous works from the authors (including the one attached as a supplementary file). Please, also add a final paragraph explaining the organization of the manuscript.

Specific Comment 5

Section 2.1

Please present the main properties of each constituent of concrete and also a table with the mix design (including units). Also, what are the main properties of the used reinforcement rebars?

Specific Comment 6

Section 2.1

Elastic Modulus of 50 GPa for a current concrete seems to be very high. Please clarify!

Also, in Table 1, in most current standards strength should be referred with “f” and not “sigma”.

Specific Comment 7

Tables 1, 2 and 3

Please, explain how the velocity of elastic wave in Table 1 and the values presented in Table 3 were measured. Please, also present the main properties of fine powder and binder from Table 2.

Specific Comment 8

Captions in figures

Avoid to write additional comments in the captions in figures. For instance, in Figure 1, the sentence “Drilling process … one [10]” should be deleted. Same for other figures.

Also, in Figure 3, sub captions for Figures 3(a), (b) and (c) are missing.

Specific Comment 9

Section 2.2

This section present a mathematical framework which is not validated against experimental results. Please, remove this section or validate the predictions from the equations with your experimental results. This can imply to reorganize the manuscript. Also, all well-established equations which are not derived here must be supported with a reference (also valid for other sections which incorporate equations).

Specific Comment 10

All tables, figures and equations must appear after they were cited in the text. As example, a new section should not start with a figure. Please refer to the guide for authors.

Specific Comment 11

Section 2.3

Please comment Figure 4 in the manuscript.

Specific Comment 12

Please explain and comment better Figure 9 in the manuscript.

Specific Comment 13

Please explain and comment better Figure 10 c) in the manuscript.

Specific Comment 14

In the references list, some of them are incomplete. Please correct them.

Author Response

2023.1.28

Dear Reviewer, 3

    Thank you for your valuable opinions.  I answered them one by one. Please check them.

Reviewer 3.

General Comment

Following the previous works from the authors related with a developed novel dismantling process for reinforced concrete using steam pressure cracking system, this manuscript mainly focusses on the examination of the drilling process, on a method to observed the fractured concrete surfaces and a study based on the specific energy consumption measured during the drilling and crack propagation to estimate the required materials and time. For this, an experimental campaign was performed. The results are presented and discussed.

The topic of the manuscript, as well as the developed novel dismantling process, is very interesting and could be used in the next future as an alternative more safer and more environment friendly dismantling technique.

I consider that in the present form the manuscript is not acceptable for publication. I made some comments in order to improve the manuscript. The authors should take the comments into account and revise their manuscript.

Specific Comment 1

The manuscript needs an extensive revision to correct the reading, tense, spelling, grammar and typos. Avoid to write in the first person of plural. Several sentences are not complete and hard to understand (the manuscript is not easy to read), and several formatting issues (including symbols and numbering of figures, among others) need also to be corrected. The authors should seek to professional help for this revision and also refer to the guide for authors. I don’t present examples because they are so many.

Answer 1: I apologize that my inadequate manuscript confused you. I asked to professional help and corrected as possible as I could.    Thanks for pointing it out.

Specific Comment 2

Abstract

The abstract must be revised. It must summarizes the achievements of the previous works from the authors, and highlight the novelty and importance of this manuscript. In addition, the main results of this research should be also summarized.

Answer 2: I have completely revised the abstract (line 17-29 in revised manuscript) according to the reviewer comment. Thank you very much for useful comment.

Specific Comment 3

Keywords

Additional keywords must be added and refer to the novelty of this study.

 Answer 3: The author revised the keywords as follows:  Keywords:. quick dismantling, reinforced concrete, steam pressure cracking, less vibration, remote system, non-pollution.”

Specific Comment 4

Introduction

The introduction section is somewhat poor and to short. The literature review must be updated and improved. In a more detailed way, in the end of the introduction the novelty and importance of this study must be clearly explained, in relation with the previous works from the authors (including the one attached as a supplementary file). Please, also add a final paragraph explaining the organization of the manuscript.

Answer 4:  The author enriched the introduction (line 33-103 in revised text) by adding the novelty of this research.

The final paragraph explained the organization as follows: “This research was conducted based on an official tripartite agreement with Akita University,  Nippon Koki Co. Ltd. and SANWA TEKKI Corporation.”

Specific Comment 5

Section 2.1

Please present the main properties of each constituent of concrete and also a table with the mix design (including units). Also, what are the main properties of the used reinforcement rebars?

Answer 5:  Author added follwing explanetion to clear the main properties. (line 108-111)

“As mentioned in the Introduction, the strength of concrete varies depending on the mixing ratio and curing time. In this study, assuming high-strength concrete for a reactor structure, the compressive strength of a large test piece (1m×1m×0.5 m) was adjusted to approximately 50 MPa..”

   I also explained about the steel rebars in text (line 113-114) as follows; “The steel rebars used were SD345  (JIS: Japanese Industrial Standard) with tensile strength of 490 MPa.”

Specific Comment 6

Section 2.1

Elastic Modulus of 50 GPa for a current concrete seems to be very high. Please clarify!

Also, in Table 1, in most current standards strength should be referred with “f” and not “sigma”.

Answer 6 in Table 1: Thank you very much for your comments.  I also check them and would like to collect as follows: Elastic Modulus has been corrected to 35 GPa and  “σ” has  been changed to ” F” in Table 1.

Specific Comment 7

Tables 1, 2 and 3

Please, explain how the velocity of elastic wave in Table 1 and the values presented in Table 3 were measured. Please, also present the main properties of fine powder and binder from Table 2.

Answer 7:  Author measured the elastic wave velocity by using a impact method in a previous study. Basically, velocity is found by dividing the propagation distance by the propagation time of elastic wave. I added a reference [26] in this table.

Fine powder is “spherical Aluminum particles with several microns”  and the binder is “metallic soaps such as sodium stearate”. I added this information in Table 2.

Specific Comment 8

Captions in figures

Avoid to write additional comments in the captions in figures. For instance, in Figure 1, the sentence “Drilling process … one [10]” should be deleted. Same for other figures.

Also, in Figure 3, sub captions for Figures 3(a), (b) and (c) are missing.

Answer 8:  The author avoids additional comments as possible. I also left sub comments if necessary to understand that Figure.

Specific Comment 9

Section 2.2

This section present a mathematical framework which is not validated against experimental results. Please, remove this section or validate the predictions from the equations with your experimental results. This can imply to reorganize the manuscript. Also, all well-established equations which are not derived here must be supported with a reference (also valid for other sections which incorporate equations).

 Answer 9 I understand reviewer comment and removed section 2.2 all.

Specific Comment 10

All tables, figures and equations must appear after they were cited in the text. As example, a new section should not start with a figure. Please refer to the guide for authors.

 Answer 10:  I appreciate the reviewer comments. I corrected the position of figures after citing in the text.

Specific Comment 11

Section 2.3

Please comment Figure 4 in the manuscript.

Answer 11. I explained Figure 2 ( Figure 4 before revision)  as follows: “The stereolithography( STL) data measured by the 3D scanner were analysed using the application  FlashPrint (Flashforge Corporation) as shown in Figure 2.  The analysis method of roughness at the cracked concreate surface was as follows:

(1) Scanning the concrete surface using a precise 3-D scanner (Revopint 3D)

(2) Changing the scanning date to STL data using scanner software.

(3) Reading STL data using a 3D-printing software (FlashPrint) .

(4) Analysing the roughness data (z-direction) from the 3D coordinate data in Figure 2..”   

Specific Comment 12

Please explain and comment better Figure 9 in the manuscript.

Answer 12: Thank you so much for pointing out my poor explanation of Figure 8 ( original Figure 9 became Figure 8 in revised manuscript) as follows ( Line 245-254);  “The relationship between the distance of the drill tip from the surface and drilling time while the SPC system was operating is shown in Figure 8.  The straight line from the origin indicates that the tip of the bit contacted the surface of the concrete and began drilling. The gradient of the line indicates the drilling speed of concrete cutting.  The gradient suddenly changed at a drilling distance of 35 mm, indicating that the drill tip reached the reinforced steel bar.  The drilling speeds of concrete and steel were 0.61 and 0.0175 mm/s, respectively. The drilling speed recovered at a drilling distance of 45 mm, because the steel bar of 9 mm diameter was drilled through and the concrete was being drilled again. “

Specific Comment 13

Please explain and comment better Figure 10 c) in the manuscript.

Answer 13 I explain about Figure 9(c) ( original Figure 10 became Figure 9 in revised manuscript) as follows; (line 273-278)

“The variation in the roughness shown in Figure 9(c) can be related to the difference in the crack propagation speed.   Generally, the failure mode of the material exhibits a tendency to brittle fracturing with an impact load compared with a static load. The impact pressure owing to the SPC was the highest near the centre, and the pressure decreased as it moved away. That is, the closer to the centre, the more the material become brittle, and the cracks propagate brittle and flatly.

Specific Comment 14

In the references list, some of them are incomplete. Please correct them.

 Answer 14: I have checked carefully and corrected them.

Submission Date

18 December 2022

Date of this review

08 Jan 2023 11:35:04

Answer date 2023.1.28

Osamu Kamiya

Round 2

Reviewer 1 Report

Dear Authors,

you have taken the article to a new level. 

Reviewer 2 Report

After taking into account the reviewer's suggestions, as well as after making extensive additions to the text and adding new figures, I consider the manuscript suitable for publication in its present form.

Reviewer 3 Report

I received and read the revised version of the article “Dismantling of Reinforced Concrete Using Steam Pressure Cracking System: Drilling and Crack Propagation”. The authors have improved the manuscript according to my previous comments. I consider that the manuscript have reached the minimum quality to be accepted for publication.